Hypoxia-inducible C-to-U coding RNA editing downregulates SDHB in monocytes

Baysal Bora E. 1 bora.baysal@roswellpark.org
De Jong Kitty 1
Liu Biao 2
Wang Jianmin 2
Patnaik Santosh K. 3
Wallace Paul K. 1
Taggart Robert T. 1
1 Departments of Pathology and Laboratory Medicine, Roswell Park Cancer Institute , Buffalo, NY , USA
2 Biostatistics and Bioinformatics, Roswell Park Cancer Institute , Buffalo, NY , USA
3 Thoracic Surgery, Roswell Park Cancer Institute , Buffalo, NY , USA
Young Howard
Electronic publication date: 2013 Sep 10
Publication date: 2013
Volume: 1
Electronic Location ID: e152
Received 2013 Jun 13; Accepted 2013 Aug 14
Copyright: © 2013 Baysal et al.
Copyright year: 2013
Copyright holder: Baysal et al.
License: This is an open access article distributed under the terms of the Creative Commons Attribution License, which permits unrestricted use, distribution, and reproduction in any medium, provided the original author and source are credited.
License URL: https://creativecommons.org/licenses/by/3.0/

Keywords: Epigenetic, Environment, RNA editing, Cytidine deaminase, Monocyte, Macrophage, Mitochondrion, Hypoxia, Complex II

Funding: Roswell Park Cancer Institute’s Flow & Image Cytometry Resource NIH Shared Instrument Program Cancer Center Support Grant 5 P30 CA016056-29 National Cancer Institute to the Roswell Park Cancer Institute This work is supported by RPCI departmental startup funds to BEB. Flow cytometry was performed at Roswell Park Cancer Institute’s Flow & Image Cytometry Resource, which was established in part by equipment grants from the NIH Shared Instrument Program, and receives support from a Cancer Center Support Grant (5 P30 CA016056-29) from the National Cancer Institute to the Roswell Park Cancer Institute. The funders had no role in study design, data collection and analysis, decision to publish, or preparation of the manuscript.

==============================
Background. RNA editing is a post-transcriptional regulatory mechanism that can alter the coding sequences of certain genes in response to physiological demands. We previously identified C-to-U RNA editing (C136U, R46X) which inactivates a small fraction of succinate dehydrogenase (SDH; mitochondrial complex II) subunit B gene (SDHB) mRNAs in normal steady-state peripheral blood mononuclear cells (PBMCs). SDH is a heterotetrameric tumor suppressor complex which when mutated causes paraganglioma tumors that are characterized by constitutive activation of hypoxia inducible pathways. Here, we studied regulation, extent and cell type origin of SDHB RNA editing.

Methods. We used short-term cultured PBMCs obtained from random healthy platelet donors, performed monocyte enrichment by cold aggregation, employed a novel allele-specific quantitative PCR method, flow cytometry, immunologic cell separation, gene expression microarray, database analysis and high-throughput RNA sequencing.

Results. While the editing rate is low in uncultured monocyte-enriched PBMCs (average rate 2.0%, range 0.4%–6.3%, n = 42), it is markedly upregulated upon exposure to 1% oxygen tension (average rate 18.2%, range 2.8%–49.4%, n = 14) and during normoxic macrophage differentiation in the presence of serum (average rate 10.1%, range 2.7%–18.8%, n = 17). The normoxic induction of SDHB RNA editing was associated with the development of dense adherent aggregates of monocytes in culture. CD14-positive monocyte isolation increased the percentages of C136U transcripts by 1.25-fold in normoxic cultures (n = 5) and 1.68-fold in hypoxic cultures (n = 4). CD14-negative lymphocytes showed no evidence of SDHB editing. The SDHB genomic DNA remained wild-type during increased RNA editing. Microarray analysis showed expression changes in wound healing and immune response pathway genes as the editing rate increased in normoxic cultures. High-throughput sequencing of SDHB and SDHD transcripts confirmed the induction of C136U RNA editing in normoxic cultures but showed no additional verifiable coding edits. Analysis of SDHB RNA sequence data from 16 normal human tissues from the Illumina Body Map and from 45 samples representing 23 different cell types from the ENCODE projects confirmed the occurrence of site-specific C136U editing in whole blood (1.7%) and two primary CD14+ monocyte samples (1.9% and 2.6%). In contrast, the other cell types showed an average of 0.2% and 0.1% C136U editing rates in the two databases, respectively.

Conclusions. These findings demonstrate that C-to-U coding RNA editing of certain genes is dynamically induced by physiologically relevant environmental factors and suggest that epigenetic downregulation of SDHB by site-specific RNA editing plays a role in hypoxia adaptation in monocytes.

Introduction

RNA editing is a post-transcriptional regulatory mechanism which often results in conversion of adenosine to inosine (A-to-I) in mRNA sequences (Nishikura, 2010). Recent whole transcriptome sequence analyses reveal abundant A-to-I type RNA editing in non-coding and Alu-containing transcript regions while RNA editing of protein encoding regions, especially of non-A-to-I types, appears to be very rare (Kleinman, Adoue & Majewski, 2012; Piskol et al., 2013). Site-specific C-to-U RNA editing leading to amino acid recoding of an endogenous gene in normal mammalian cells was previously confirmed, to our knowledge, only in the ApoB gene encoding apolipoprotein B (Blanc & Davidson, 2003). C-to-U editing of Apo B is catalyzed by APOBEC1 cytidine deaminase that generates a shorter protein isoform in intestinal epithelial cells. APOBEC1 also inactivates 4–17% of neurofibromatosis type 1 (NF1) gene transcripts by site-specific C-to-U RNA editing in certain high-grade NF1 tumors but not in normal cells (Skuse et al., 1996).

We previously identified SDHB C-to-U mRNA editing (C136U) at low steady-state levels in peripheral blood mononuclear cells (PBMCs) of normal individuals using reverse transcription (RT) and semi-quantitative PCR (Baysal, 2007). Analysis of the purified PBMC subsets obtained from a single donor showed that the C136U editing rate was higher in monocytes than in lymphocytes. SDH is a central metabolic enzyme in Krebs cycle (Rutter, Winge & Schiffman, 2010) and a tumor suppressor for hereditary paraganglioma (PGL) (Baysal et al., 2000). SDH catalyzes the oxidation of succinate to fumarate during aerobic respiration. In anaerobically respiring mitochondria of lower organisms, SDH is inactive and fumarate is often reduced to succinate in a reverse reaction catalyzed by fumarate reductase (Muller et al., 2012). Germline inactivating mutations in the nuclear-encoded SDH subunit genes, primarily in SDHB and SDHD, cause PGL tumors (Burnichon et al., 2012) which show constitutive activation of the hypoxia-inducible pathways. PGL tumors recapitulate the high-altitude associated carotid body (CB) paragangliomas (Saldana, Salem & Travezan, 1973), show increased severity with increased altitudes (Astrom et al., 2003; Cerecer-Gil et al., 2010) and share transcriptome characteristics with Von-Hippel Lindau (VHL) disease tumors (Dahia et al., 2005; Lopez-Jimenez et al., 2010). The VHL gene product plays an important role in normoxic degradation of hypoxia-inducible factors (HIFs) (Kaelin & Ratcliffe, 2008). Succinate and reactive oxygen species that accumulate upon SDH inactivation were implicated as downstream messengers leading to stabilization of HIF1α in normoxic conditions (Selak et al., 2005; Guzy et al., 2008). However, mechanisms of PGL tumorigenesis remains unconfirmed partly because animal or cell culture models that link SDH mutations and hypoxia are lacking.

The SDHB C136U RNA editing converts a highly conserved arginine residue to a premature stop codon (R46X) shortly after the mitochondrial targeting signal and inactivates the SDHB gene product, the iron-sulfur subunit of the SDH complex. The R46X germ line mutation was previously described in multiple index patients with PGL confirming its pathogenicity (Bayley, Devilee & Taschner, 2005). Here, we studied the SDHB C136U transcript editing in short-term cultured monocyte-enriched PBMCs using a novel allele specific quantitative PCR assay (AS qPCR), high-throughput genetic methods, immunophenotyping, immunoseparation, morphology and database analysis. Our aim was to determine the prevalence, distribution, cell type origin and other factors influencing SDHB editing using a reproducible assay to gain functional insights. We found that SDHB C136U editing is present at low levels in fresh uncultured total and monocyte-enriched PBMCs, but it markedly increases during normoxic macrophage differentiation in vitro and upon short-term hypoxic exposure in monocytes. These results provide unprecedented evidence that functional transcript dosage for a central metabolic enzyme is regulated by environmentally-inducible site-specific C-to-U RNA editing.

Materials and Methods

Leukocyte isolation and cell culture

Leukocytes were isolated from Trima leukoreduction filters (Terumo BCT, Lakewood, CO) of anonymous healthy platelet donors following an IRB-approved protocol. PBMCs were purified by histopaque 1077 (Sigma-Aldrich, St. Louis, MO) and washed twice with RPMI-1640/10% fetal bovine serum (FBS) to remove residual platelets. Each filter frequently gave 5 × 108 or more PBMCs. Monocytes were enriched by the cold aggregation method (Mentzer et al., 1986) with certain modifications. PBMCs were suspended in 35 ml RPMI-1640 in a 50 ml polypropylene tube and incubated at 4°C for 1 h on a rocker panel for homotypic aggregation of monocytes. For monocyte enrichment, the tube was positioned upright, underlaid with 6 ml FBS and incubated overnight at 4°C. The cell precipitate under the serum was collected for culture. Monocytes enrichment was approximately 70% (n = 5) by short-term precipitation (up to 3 h) and 27% (n = 10) by overnight precipitation over serum as determined by percentages of the CD14+ cells by flow cytometry. The non-aggregating upper layer was markedly depleted of monocytes (less than 4%, n = 3). We used the overnight precipitation method which gave higher yield of total cells and led to more consistent development of adherent aggregates that were associated with higher levels of C136U editing in culture. Precipitated cells under the serum were centrifuged for 10 min at 250×g and suspended in 20 ml RPMI-1640 with 10% FBS and penicillin/streptomycin. Cell density was calculated by a hemocytometer (Neubauer improved, InCyto, Covington, GA). After further dilution, two milliliters of the cell suspension was distributed to each well of a six-well tissue culture plate (Costar, Corning Incorporated, Corning, NY). Neutrophils were isolated as described (Maqbool et al., 2011) from the precipitated cell fraction containing red cells after histopaque 1077 centrifugation of the leukoreduction filter cells. Giemsa staining showed that approximately 80% of the isolated cells were granulocytes and about 20% were lymphocytes. Editing rates were determined in daily collected individual wells. We cultured 21–30 × 106 cells/2 ml per well (also see results). Cultures were incubated at 5% CO2 with either 21% O2 (normoxia) or 1% O2 combined with 94% nitrogen (hypoxia). Hypoxic cultures were pre-incubated at 37°C, 21% O2 for 2–3 h before placing them in the hypoxia chamber (XVIVO system, BioSpherix). Fresh culture media was added after 7 days in culture.

Lymphoblastoid cell line RNAs were isolated from previously described EBV-transformed PBMCs (Baysal, 2007; Baysal et al., 2000). HEK 293T embryonic kidney cell line and THP-1 monocytic leukemia cell line was purchased from ATCC (Manassas, VA). Various cytokines/differentiating agents were added for the following working concentrations: m-CSF (50 ng/ml), GM-CSF (50 ng/ml), IL4 (50 ng/ml, vitamin D (10 nM), retinoic acid (1 µM), PMA (4 beta-phorbol 12-myristate 13-acetate) (100 nM). Cytokines were purchased from Peprotech (Rocky Hill, NJ) and differentiating agents were purchased from Sigma-Aldrich. Lipopolysaccharide (List Biological Libraries Inc, Campbell, CA) is used at a final concentration of 100 ng/ml.

Microscopy and imaging

Standard tissue culture monitoring and imaging was performed by Zeiss Axio microscope. Adherent aggregates (approximately 100 µm or larger in diameter) were counted at low power (2.5×) magnification. Live image photographs were taken using a Leica AF6000LX system (Houston, TX) which is comprised of a Leica DMI-6000B microscope and Leica LAS AF software interface.

Nucleic acid isolation and analysis

RNA and DNA were isolated using Trizol (Life Technologies, Grand Island, NY) and DNA Wizard genomic DNA purification kit (Promega, Madiosn, WI), respectively. Nucleic acid lysis buffer or Trizol was added directly to the adherent cells in plate well. Nucleic acid was quantified by a spectrophotometer (Nanondrop, Thermo Scientific, Wilmington, DE). Oligonucleotide primers and templates were obtained from IDT technologies (Coralville, IA).

RT and qPCR were performed following the manufacturer’s kits and instructions (LightCycler 480II, Roche, Indianapolis, IN). Total RNA was synthesized using a mixture of random short oligonucleotide and oligodT primer mixture. PCR control experiments were performed to test the specificity and amplification efficiency of oligonucleotide primer pairs using synthetic oligonucleotide templates. qPCR amplification of cDNAs was performed and crossing point (Cp) levels of C136U (T assay) and total SDHB (total assay) were determined separately. Each assay was performed in duplicate or triplicate wells. T assay and total assay had the same forward oligonucleotide primer in exon 1 and different oligonucleotide reverse primers in exon 2, where the C136U editing occurs. The reverse primer for the T assay had an extra T at its 3′-end relative to the reverse primer in the common assay. (Oligonucleotide primers and probes used in this study are listed in Table S1.)

The relative amount of the edited transcripts was calculated by the delta-Cp method by exponentiating the total transcript Cp minus the edited transcript Cp value to the power of 2. This approach assumes 100% duplication efficiency per cycle in both assays. Control PCR reactions were performed to test the specificity and amplification efficiency of oligonucleotide primer pairs using synthetic oligonucleotide templates. Assays with 8-fold serially diluted synthetic oligonucleotide templates for the wild-type and C136T edited sequences showed a duplication efficiency of 107% for both total and T assays between 1 femtomole and 0.245 attomole (Table S2). The mutation specific primer had a high specificity for the edited transcripts, with an average false positive amplification rate of 1.38% when the control wild type oligonucleotide template sequence was amplified in an over 25,000 fold concentration range. When 100% T template was used, the average estimate of the C136U edited transcripts was 91% between 8 femtomoles and 0.245 attomoles concentration range. These control experiments suggest that AS qPCR estimates may be slightly lower than the true editing rates. We chose this stringent AS qPCR method to ensure we did not overestimate the C136U editing rates. The estimated percentage of edited transcripts using this method was highly reproducible in biological samples. For example, a set of 14 samples containing both low and high levels of edited transcripts gave very similar results in two replicate experiments with a Pearson correlation coefficient of 99.4%. In each assay, a positive control sample which had a mutation rate of 15%–17% was included. Relative quantification of CDA and SDHB expression was performed by 2−ΔΔCT method (Livak & Schmittgen, 2001) using beta-2 microglobulin as the control housekeeping gene.

Microarray analysis of gene expression

Expression profiling was accomplished using the Human HT-12 whole-genome gene expression array and direct hybridization assay (Illumina, Inc.). Initially, 500 ng total RNA was converted to cDNA, followed by in vitro transcription to generate biotin labeled cRNA using the Ambion Illumina TotalPrep RNA Amplification Kit (Ambion, Inc.) as per manufacturer’s instructions. 750 ng of the labeled probes were then mixed with hybridization reagents and hybridized overnight at 58°C to the HT-12v4 BeadChips. Following washing and staining with Cy3-streptavidin conjugate, the BeadChips were imaged using the Illumina iScan Reader to measure fluorescence intensity at each probe. The intensity of the signal corresponds to the quantity of the respective mRNA in the original sample.

The background corrected gene expression levels were extracted from BeadChip using Illumina’s Genome Studio (v2011.1) gene expression module (v1.9.0). The log2 transformed expression levels were quantile normalized using Lumi module in the R-based Bioconductor package (Gentleman et al., 2004). For data quality control, we excluded the genes with detection p-value greater than 0.05 (i.e., indistinguishable from the background noise). 18941 out of 34686 genes passed this filtering for downstream analysis.

The Limma program (Smyth, 2004) was used to calculate the level of differential gene expression. Briefly, a linear model was fit to the data (paired design, with cell means corresponding to the different condition and a random effect for array) and selected contrasts of condition (i.e., case vs. control) were performed. A list of differentially expressed genes with P < 0.05 and ≥ 2 fold-change was obtained and analyzed for enriched Gene Ontology (GO) categories and KEGG pathways using NCBI DAVID Bioinformatics Resources (Huang da, Sherman & Lempicki, 2009). The enriched GO terms and KEGG pathways with P < 0.05 and ≥ 5 genes were kept.

Accession number

Microarray data have been deposited in GEO (www.ncbi.nlm.nih.gov/geo/) under accession number GSE45900.

High-throughput RT-PCR amplicon sequencing

SDHB and SDHD gene transcripts were amplified with oligonucleotide primers derived from 5′- and 3′-UTR using Roche reverse transcriptase and Pfu Ultra2 (Agilent, Santa Clara, CA) high fidelity PCR amplification. Library preparation was performed using the Nextera XT DNA sample preparation kit (Illumina, San Diego, CA). High-throughput sequencing is performed on a MiSeq personal sequencing platform.

Flow cytometry

Cells (0.5–1 × 106) were pelleted by centrifugation at 400 × g for 5 min and incubated for 20 min at room temperature in a total volume of 100 µl with a titrated cocktail of APC-conjugated CD14 (Invitrogen, Carlsbad, CA), Brilliant-violet 421 conjugated CD16 (Biolegend, San Diego, CA), PE-conjugated CD163 (Trillium Diagnostics, Brewer, ME), FITC-conjugated CD206 (BD Biosciences, San Jose, CA), and PE-Cy7-conjugated HLA-DR (BD Biosciences, San Jose, CA), followed by a wash in PBS and resuspension in 500 µl PBS. Flow cytometric acquisition was performed on a FACS Aria I flow cytometer (BD Biosciences) equipped with four laser excitation sources (405 nm 30 mW; 488 nm 15 mW; 561 nm 30 mW; and 631 nm 17 mW) that was quality-controlled on a daily basis by using CS&T beads and FACS DiVa software (BD Biosciences). The filter configurations for the PMTs measuring fluorescence emission of the applied fluorochromes were 450/50 nm (Brilliant-violet 421); 530/30 nm (FITC); 582/15 nm (PE); 780/60 nm (PE-Cy7); and 660/20 nm (APC). Autofluorescence and single-color controls were acquired to perform spectral overlap compensation using the automated compensation matrix feature in FACS DiVa software. A forward scatter threshold was applied to eliminate electronic noise and small particles from the flow cytometric acquisition. A target of 20,000 scatter-inclusive events was acquired for each specimen. Data analysis was performed with FlowJo software version 10.0.4 (Tree Star, Inc., Ashland, OR).

Isolation of CD14+ monocytes

Monocytes were isolated either by flow cytometric gating of CD14+ (both dim and strong intensity) or by CD14 microbeads (Miltenyi Biotec Inc. Auburn, CA) following manufacture’s protocol. Flow cytometry showed 93% or higher purity after CD14 microbead purification.

Immunocytochemical staining

PBMCs were grown in tissue culture chamber slides (Lab Tek Chamber slide, Fisher Scientific). Before staining, supernatant was discarded and adherent cells were washed once with PBS. Adherent cells were fixed by alcohol-based cytology fixative (Leica microsystems) and air dried. For antigen retrieval, slides were heated in the steamer for 20 min in citrate buffer (pH 6.0), followed by a 20 min cool down. Endogenous peroxidase was quenched with aqueous 3% H2O2 for 10 min and washed with PBS/T. Slides were loaded on a DAKO autostainer and a serum free protein block (Dako catalog #X0909) was applied for 5 min, blown off, and the antibody applied for one-hour. Biotinylated goat anti-mouse IgG (Jackson Immuno Research Labs, catalog #115-065-062) was applied for 30 min, followed by the Elite ABC Kit (Vectastain) for 30 min, and the DAB chromagen (Dako) for 5 min. Finally, the slides were counterstained with hematoxylin, dehydrated, cleared and cover slipped.

Western blot analysis

Cell lysates were prepared using 75 µl of M-PER protein extraction reagent (Thermo, Waltham, MA) per approximately 20 × 106 washed cells. The concentration of proteins in lysates was determined with the colorimetric Pierce 660 nm protein assay (Thermo).

For electrophoresis, lysates (40–60 µg of protein) were boiled for 5 min in buffer containing 50 mM Tris, 2% SDS and 143 mm β-mercaptoethanol. After electrophoresis in Mini-PROTEAN™ TGX pre-cast 4%–15% gradient polyacrylamide gels (Bio-Rad, Hercules, CA) at 15–30 mA for 90 min, proteins were transferred overnight at 30 mA in 10% methanol-containing Tris-glycine buffer to 0.45 µ polyvinylidene difluoride membrane (GE Healthcare, Little Chalfont, UK).

For immunoblotting, membranes were blocked in Tris-buffered saline (TBS; 10 mM Tris-HCl and 150 mM NaCl at pH 7.4) with 0.05% Tween-20 (Sigma, Saint Louis, MO) and 5% non-fat milk (Carnation™, Nestle), and then incubated in the same solution with a primary mouse monoclonal anti-SDHB antibody (sc-271548, Santa Cruz Biotechnology, Dallas, TX) at room temperature for an hour. After washing in TBS/Tween-20 for 30 min, membranes were incubated in TBS/Tween-20/milk with a horseradish peroxidase-conjugated secondary antibody for 1 h at room temperature. Membranes were then washed in TBS/Tween-20 for 30 min, incubated in Luminata Forte™ chemiluminescence reagent (Thermo), and exposed to photographic film (Thermo).

Mouse anti-beta-actin (clone C4, Santa Cruz Biotechnology) and horseradish peroxidase-conjugated goat anti-mouse IgG (Invitrogen, Carlsbad, CA) antibodies were used at 1:200 and 1:2500 dilutions, respectively. Band intensities were determined by a densitometer. the average of two density measurements from beta actin was used to determine relative amounts of SDHB protein.

RNA sequence database analysis

BAM sequence alignment files of RNA-seq reads mapped against the GRCh37/hg19 human genome assembly were downloaded from public data repositories of the Illumina® BodyMap 2.03 and ENCODE human transcriptome projects (Derrien et al., 2012; Tilgner et al., 2012). The two projects have analyzed poly-A + RNA using the Illumina® HiSeq 2000 or Genome Analyzer II or IIx RNA sequencing platforms. From the BodyMap project, data for all 16 tissues from different healthy individuals was obtained. From the ENCODE project, data for 23 different cell-types/cell-lines (22 with biological duplicates) was obtained.

For each BAM file, the mpileup routine in samtools7 0.1.19 (March 2013 release) was used with its default setting, (Li et al., 2009) which weighs alignment quality and ignores duplicate reads, to identify RNA-seq reads that align against chromosome 1 regions for the 843 b-long coding region of human SDHB mRNA (NCBI RefSeq8 no. NM_003000.2). The output of mpileup was then scanned with a Python 2.7 script to quantify the number of RNA-seq reads with one of five base calls (A, C, G, T or ambiguous) at each nucleotide position. The fraction of RNA-seq reads with a base call of T for c.136C was compared against the fraction with a base call of T at any of the 213 C nucleotide-bearing positions of the SDHB coding region. Statistical significance of any difference was determined with two-tailed Chi-square test with Yates correction using Prism 6.0c software (GraphPad®).

None of the 67 single nucleotide polymorphisms (SNPs) that have been documented for the SDHB coding region in dbSNP10 database (04-25-2012 release, NCBI dbSNP build 138 phase I) is at the genomic position of SDHB c.136C.

Statistical analysis

Unless specified otherwise, statistical analyses of experimental data was performed using the online interface of SISA at http://www.quantitativeskills.com/sisa (Uitenbroek, 2013) using two-tailed non-parametric tests. Graphics and descriptive statistics were generated in Excel (version 10; Microsoft, Redmond, WA). Means and associated standard errors are depicted by horizontal bars in the figures.

Results and Discussion

Analysis of C136U mutation rate in PBMCs by RT and AS qPCR

To determine the prevalence, distribution, cell type origin and other factors influencing SDHB editing, we developed a highly specific and reproducible assay for allele specific quantitative PCR amplification (AS qPCR) of the total and edited C136U SDHB mRNAs (see methods). Using this assay, we found that the C136U transcripts were very low or absent in B cell lymphoblastoid and embryonic kidney cell lines. The editing occurred at slightly higher levels in PBMCs isolated from fresh blood than in the cell lines (Fig. 1A). To study the relative contribution of monocytes and lymphocytes to SDHB editing, we performed monocyte enrichment using the well-established cold aggregation method (Mentzer et al., 1986) with certain modifications (see methods). This method essentially separates PBMCs into cold-aggregating monocyte-enriched and non-aggregating monocyte-depleted (i.e., predominantly lymphocytic) compartments. Monocyte-enriched samples by the cold aggregation method showed higher editing rates than the matched PBMCs (2.16% versus 1.48%, n = 36, p = 1.6 × 10−5, Wilcoxon matched pairs signed ranks test) and than the matched monocyte-depleted samples (1.47% versus 0.58%, n = 10, p = 0.005, Wilcoxon matched pairs signed ranks test) (Fig. 1A). To further confirm monocyte-origin of the edited transcripts, we isolated peripheral blood monocytes to >92% purity by CD14+ microbeads and tested the mutation rates in three samples. In each sample, a higher mutation rate was found in the monocytes than the CD14− lymphocytes (average 0.84% vs 0.38%). A positive but weak correlation was found between the CD14+ monocyte-fraction and the C136U editing rate in cold-aggregated PBMC samples (Fig. 1B). A linear regression model provided a point estimate of 4.25% C136U editing rate in a pure monocytic cold-aggregated population. These results indicate that C136U editing occurs in freshly isolated monocytes in low but statistically significantly higher rates than in lymphocytes.

Figure 1 SDHB C136U RNA editing is induced in monocytes during short-term culture.

(A) C136U editing is measured in 9 lymphoblastoid cell lines and one embryonic kidney cell line, freshly isolated PBMCs (n = 50), PBMCs monocyte-enriched by cold-aggregation (Mono+; n = 42) and monocyte-depleted PBMCs comprised of largely lymphocytes (Mono-; n = 10). Horizontal lines represent mean ± standard errors throughout the figures. (B) Fraction of the C136U transcripts weakly correlates with the CD14+ monocyte percentage in monocyte-enriched PBMCs (Pearson correlation coefficient r = 0.36). (C) Impact of short-term culture on C136U editing. Short-term culture of five monocyte-enriched PBMCs shows upregulation of C136U editing in adherent cells in culture days 5–7 compared to days 1–3. Day 0 represents the uncultured samples. (D) Flow cytometric sorting of CD14+ versus CD14− adherent cells on culture days 5–8 shows higher mutation rates in monocyte-macrophage lineage than in lymphocytes (p = 0.04, n = 5, Wilcoxon matched pairs signed ranks test). (E) C136U editing is lower among non-adherent cells in the supernatant than in adherent cells (4.3% versus 9.8%) on culture days 5–7 (n = 19 wells from 11 samples on days 5–7, p = 2.1 × 10−4, Wilcoxon matched pairs signed ranks test).

C136U editing rate increases during short-term culture of PBMCs in a time-dependent manner

We cultured monocyte-enriched PBMCs for in vitro macrophage differentiation by plate adherence in the presence of 10% fetal bovine serum (FBS) at standard culture conditions as described (Roiniotis et al., 2009). Flow cytometry showed that the fraction of CD14+ cells among the adherent cells on culture days 5–6 (25.9% ± 5.2%; n = 4) was comparable to the uncultured monocyte-enriched PBMC samples (27.2% ± 2.7%; n = 10). Initial cultures showed that the fraction of the edited transcripts increased in the adherent cells on days 5–7 then decreased to baseline levels on later days. (Fig. S1A). Short-term cultures showed that the editing rates were lower in the first three days than on days 5–7 (Fig. 1C, 3.54% versus 11.62%, n = 15, p = 8 × 10−4, Wilcoxon matched pairs signed ranks test). The uncultured monocyte-enriched PBMC samples had lower C136U editing rates than their matched cultures on days 5 and 6 in 17 of the 17 samples (1.7% ± 0.2% versus 10.06% ± 0.8%; p = 2.9 × 10−4, Wilcoxon matched pairs signed ranks test). To confirm monocyte origin of the edited transcripts in the cultured samples, we performed flow cytometric sorting of the attached CD14+ cells from culture days 5–8 which essentially separated monocyte/macrophages from lymphocytes to >95% purity. In five of five samples, the CD14+ cells showed higher editing rates than the CD14− cells (Fig. 1D). The editing rate was higher among the adherent cells than those in the supernatant (Fig. 1E). Collectively, these findings indicate that C136U editing is induced in monocyte/macrophage lineage cells during culture of monocyte-enriched PBMCs in a time-dependent manner.

The maximum mutation rate reached on days 4–8 varied markedly among samples from 1.2% to 18.8%. Total number of cultured cells per well correlated positively but very weakly with the maximum mutation rate (Fig. S1B). The maximum C136U editing rate trended higher when the total number of monocyte-enriched PBMCs was over 21 million per well versus 20 million or less per well, though this difference did not reach statistical significance (10.9% ± 1.23%; n = 16 versus 8.54% ± 1.03%; n = 15; p = 0.1; Mann Whitney U test, one-sided).

Morphologic evaluation of cultures associated with C136U editing

We noted the development of three dimensional adherent aggregates (AAs) in monocyte-enriched PBMC cultures, especially in those that had the highest C136U editing rates. These aggregates were detectable within 12 h and increased in density and size up to a week in culture (Figs. 2A and 2B). Hourly photographic imaging in the first two days showed that the aggregates start as a loose collection of cells and become larger and more compact by merging and recruitment of individual cells (Video S1). Cultures with low AA densities showed lower mutation rates than those with intermediate or high AA densities (p = 0.015, n = 16, Mann-Whitney U test, Fig. 2C).

Figure 2 SDHB C136U RNA editing correlates with adherent monocyte-rich adherent aggregates (AA) in culture.

The AAs on culture days 1 (A) and 5 (B) are shown (5× low power field; vertical bars equal to 1 mm). The culture is initiated with 30 million cells per well. The AAs grow in size until approximately days 5–7 when the C136U editing rate also usually peaks. (C) High editing rates are seen in cultures with intermediate and high density of adherent aggregates. Aggregate densities are grouped as follows: low = less than 10 aggregates/per 5× low power field (lpf); intermediate = 10–20 aggregates/lpf; high = more than 20 aggregates/lpf (D) Giemsa stain highlights an adherent aggregate on day 6 culture. Individually attached mature macrophages have mostly round eccentric nuclei and abundant cytoplasm. Occasional multi-nucleated macrophages (arrowheads) and rare small lymphocytes (green arrows) are also present. Macrophages within the aggregate are smaller with less cytoplasm and occasionally have curved nuclei (arrows). (E) CD163 immunostaining of adherent aggregates. Immunocytochemical staining for monocyte-macrophage-specific antigen CD163, a scavenger receptor for hemoglobin-haptoglobin complex, demonstrates that the adherent aggregates are primarily comprised of monocyte/macrophage lineage cells. Individually attached macrophages outside the aggregate also stain with variable intensity (short arrows); while numerous small lymphocytes are negative (long arrows).

Morphologic examination showed that the AAs were primarily comprised of intermediate-size cells that generally had round non-convoluted nuclei and a moderate amount of cytoplasm. Occasional lymphocytes and large monocytoid cells with curved nuclei were also noted within the aggregates. In contrast, numerous individually attached macrophages, including those with spindle-like morphology or multi-nucleation, had more abundant cytoplasm compared to cells in the AAs (Fig. 2D). Immunostaining with CD163 antibodies confirmed that most cells in the AAs were aggregated monocytes/macrophages (Fig. 2E). These results suggest that C136U editing primarily occurs in aggregating monocytes that are in the process of differentiating to macrophages. It is conceivable that micro-environmental nutrient and/or oxygen depravation associated with the dense adherent aggregates of monocytes may induce SDHB RNA editing.

Microarray analysis of differentially expressed genes associated with increased C136U editing in normoxia

We performed pairwise microarray gene expression analysis of four sets of low-editing (day 3) and high-editing (day 5–8) samples cultured in normoxia (Table 1) to evaluate (a) global changes in cellular pathways; (b) whether adherent aggregates are associated with micro-environmental hypoxia; and (c) changes in the SDH subunit and cytidine deaminase family of genes. Analysis showed that 171 genes were upregulated and 123 genes were downregulated genes by at least two-fold at a statistically significant level (P < 0.05). The subset of these genes (n = 55) that showed ≥3 fold-change is shown in Table S3. The complete list is available on GEO database under accession number GSE45900. No SDH subunit or APOBEC family gene met these criteria. Cytidine deaminase (CDA) was the only gene from the cytidine deaminase family of genes that had its expression level changed significantly, with a 3.1-fold increase. qPCR analysis confirmed upregulation of CDA expression (average 3.7-fold) and little change in SDHB expression (average 0.98-fold) in the high-editing samples relative to the low-editing ones. Significant gene expression changes coordinately occurred in functionally related genes in several Gene Ontology (GO) categories (n = 159). This number decreased to 10 after Bonferroni correction for multiple testing was applied (Table S4). These 10 categories included defense to wounding, inflammatory, immune and defense responses, taxis, chemotaxis, carboxylic and organic acid transport, locomotory behavior and regulation of cell proliferation. Notably, hypoxia-response genes were not categorically different between the low and the high editing samples. These results indicate that significant gene expression changes occur in inflammatory and immune response pathways during increase in SDHB editing and that CDA is a candidate gene for enzymatic C136U deamination.

Table 1 Percentage of C136U transcripts estimated by allele specific (AS) qPCR and high-throughput amplicon sequencing in normoxic cultures.

Sample no.	Day in culture	AS qPCR	High-throughput sequencinga	
1	Day 3	2.2%	NRb	
1	Day 6	15.9%	22%	
2	Day 3	1.3%	NR	
2	Day 8	12.3%	23%	
3	Day 3	4%	NR	
3	Day 6	13.5%	15%	
4	Day 3	8.7%	11%	
4	Day 7	18.8%	26%	
Notes.

a Sequencing depth ranges between 4980 and 4986 for the reported variants.

b NR = Not reported. Variants detected by high-throughput sequencing are reported only when their frequency exceeds 10%.

Flow cytometric characterization of cultures associated with C136U editing

To evaluate monocyte macrophage maturation in PBMC culture, we performed flow cytometric characterization of the attached cells using monocyte/macrophage-associated antibodies for CD14, CD16, CD163, HLA-DR and CD206, a mannose receptor associated with M2 type macrophage polarization that promotes wound healing (Porcheray et al., 2005).

Cold-aggregated uncultured PBMCs showed a discrete and relatively uniform monocytic population that is positive for CD14, HLA-DR and CD163; but negative for CD206 and largely negative for CD16 (Fig. 3). Cultured adherent cells at day 6, which showed an increased editing rate, contained a CD14+ population comprised of larger cells with more complexity and were dim-positive for CD206, heterogeneously positive for CD14 and brightly positive for HLA-DR. CD16 and CD163 patterns were similar and showed a heterogeneous negative-to-positive range. Analysis of the adherent cells at day 15, when the editing rates are typically low and the AAs are loose (Fig. S1C), showed a mature macrophage population that had increased complexity and a more uniform antigenic profile, including homogenous positivity for CD14, CD16, CD163, HLA-DR and CD206. Taken together, these results demonstrate that high C136U editing peaks during monocyte macrophage differentiation in normoxic culture.

Figure 3 SDHB C136U RNA editing increases during monocyte-macrophage maturation.

Flow cytometric evaluation of monocyte-macrophage maturation is shown in uncultured cold aggregated PBMCs (first column), adherent cells on culture day 6 (second column) and adherent cells on culture day 15 (third column). The day 15 culture shows a relatively homogenous large (indicated by high forward scatter, FSC-A) macrophage population that has high complexity (indicated by high side scatter, SSC-A) and is positive for CD206 (mannose receptor), HLA-DR and CD163. In contrast, uncultured monocytes are largely a uniform population that is smaller with less complexity and is negative for CD206. The day 6 culture, which has 11.6% C136U editing rate, shows a CD14+ population that is brightly positive for HLA-DR, dim-positive for CD206 and negative for CD163. CD14+ monocyte-macrophage and lymphocyte populations are marked by green and red, respectively.

C136U editing in LPS, cytokine-treated PBMC cells and in monocytic leukemia cell line THP-1

When monocyte-enriched PBMC cultures were treated with m-CSF and GM-CSF/IL4 to facilitate macrophage and dendritic cell differentiation, respectively, lower editing rates were seen compared to the control cultures that contained only 10% FBS (p = 0.012, n = 8 and p = 0.04, n = 5, respectively, Wilcoxon matched pairs signed ranks test Fig. S1D). Similarly, lipopolysaccharide (LPS) treatment of monocyte-enriched PBMCs minimally but statistically significantly reduced the editing rates during the first three days of culture (0.62% in LPS-treated group versus 1.17% in controls in the total cell population [n = 4]; p = 0.013, Wilcoxon matched pairs signed ranks test). Notably, LPS and cytokine-treatment, especially with GM-CSF/IL4, also diminished adherent aggregate formation under normoxia. Next, we tested C136U editing rates in the monocytic leukemia cell line THP-1. We added macrophage differentiation agents to the media with 10% FBS including phorbol ester PMA (4 beta-phorbol 12-myristate 13-acetate), retinoic acid and vitamin D. Significant C136U editing was not identified (less than 1%) either in the untreated THP-1 cell line or after treatment with the differentiating agents. Similarly, hypoxia exposure of the THP-1 cell line showed no increase in C136U levels in days 1, 2 and 3 (less than 1%). These results suggest that C136U editing under normoxic culture conditions primarily occurs in mature monocytes within the dense adherent aggregates during differentiation to macrophages in the presence of serum and that LPS-driven monocyte activation (Rossol et al., 2011) or cytokine-driven macrophage or dendritic cell differentiation or leukemic transformation reduces the editing rates.

Induction of C136U editing by hypoxia

We evaluated the role of hypoxia on C136U editing for the following reasons. First, tissues with inflammation and infarction harboring monocyte/macrophage lineage cells are often hypoxic (Riboldi et al., 2013). Subsequently, monocyte-macrophage differentiation frequently occurs in tissues where oxygen pressure is lower than that in blood. Second, the dense adherent aggregates associated with increased SDHB editing in normoxic cultures might conceivably be associated with micro-environmental hypoxia. We observed that monocyte-enriched PBMC cultures in hypoxia did not develop the firmly adherent aggregates associated with high editing rates in the normoxic cultures and that essentially all cells were in suspension. Fraction of viable CD14+ monocytes among total cells in the hypoxic supernatants on culture day 2 was similar to the original uncultured samples as evaluated by flow cytometry (20.9% versus 20.4%, n = 4). The editing rate in the hypoxic cultures increased in the first three days with a peak on day 2, when it is typically low in the normoxic control cultures (Figs. 1C and 4A). In five of the 14 samples, C136U editing rates were higher (21%–49%) than in any normoxic culture. Hypoxia increased the fraction of C136U transcripts by 0.4% to 46% on culture day 2. The variable increase in the percentage of C136U transcripts could be caused by technical or biological factors including variable fraction of monocytes in donors, variable enrichment of monocytes by the cold-aggregation method, variation in inherent enzymatic activity of the putative editing enzyme or nonsense mediated decay pathway activation (Chang, Imam & Wilkinson, 2007).

Figure 4 Hypoxia induces SDHB C136U RNA editing.

(A) The editing rates are higher on days 1–3 in hypoxia than the average of adherent and supernatant cells in normoxic cultures (3-day average 11.2% in hypoxia versus 3.6% in normoxia; p = 0.006, n = 4, Wilcoxon matched pairs signed ranks test). In contrast, as the hypoxic editing rate decreases along with extensive cell death on days 5–7, it increases in normoxic cultures as previously described (Fig. 1C). The average of four independent cultures is presented. (B) C136U editing rates in 14 independent cultures in hypoxia versus normoxia during 3 days of culture are shown. The editing rates were obtained from supernatant in all hypoxic cultures (first column) and from total cell population in normoxic cultures, except in two normoxic cultures where only supernatant was collected (second column). D0 samples represent the original uncultured cold aggregated PBMCs. Hypoxia statistically significantly increased the C136U editing rates in days 1, 2 and 3. (C) PCR amplification and Taq1 restriction enzyme (RE) digestion shows no evidence of C136T DNA mutation in SDHB exon 2 genomic DNA (gDNA) (right panel) in the same samples with high C136U RNA editing rates (left panel). The C136U RNA editing rates above the lanes were measured by AS qPCR. RT- and gDNA-PCRs generate amplicon sizes of 285 bp and 233 bp, respectively. Taq1 RE digestion of wild type cDNA and gDNA sequences generates 159 bp/126 bp and 131 bp/102 bp bands, respectively. C136U/T mutation destroys the RE site. (D) Western blot analysis of normoxic (N) and hypoxic (H) cultures from one donor shows no major changes in SDHB protein expression, normalized against beta actin, as the editing rate increases in hypoxia, but possibly a subtle decrease when the C136U percentage is 16.1% on day 2 in hypoxia. (E) The editing rates are higher in flow sorted CD14+ viable cells than CD14− viable cells from day 2 hypoxic cultures (average 9.35% versus 1.63%; p < 0.04, n = 4, Wilcoxon matched pairs signed ranks test, one-sided). In all experiments, hypoxic and control normoxic cultures derive from the same donors (i.e., oxygen tension is the only variable).

Pairwise comparison of the editing rates in supernatant cells in hypoxia versus total cells in normoxia from 14 different samples showed marked upregulation of C136U editing by hypoxia in the first three days (p = 2 × 10−7, Wilcoxon matched pairs test; Fig. 4B). Hypoxia increased C136U fractions in 39 of 42 pairwise comparisons from 14 samples cultured in normoxia and hypoxia for three days. Comparison of the flow sorted CD14+ and CD14− cell populations confirmed that monocytes were the main source of C136U editing in hypoxia (Fig. 4E). Hypoxic culture of granulocytes obtained from two donors, plated at 0.6 and 1.2 million cells/ml densities, respectively, showed no evidence of C136U editing (0.89% in normoxic and hypoxic granulocytes; n = 2). In contrast, monocyte-enriched PBMCs from the same donors showed very high editing rates (>45%) under hypoxia. These findings indicate that hypoxia-induced C136U SDHB RNA editing occurs primarily in monocytes among peripheral blood leukocytes.

RT-qPCR analysis showed marked upregulation of CDA (average 7.5-fold change on day 2) and downregulation of SDHB (average 0.37-fold change on day 3) in the hypoxic cultures relative to the control gene beta 2 microglobulin. Western blot analysis of one monocyte-enriched PBMC sample cultured in normoxic and hypoxic conditions showed no marked changes in the expression SDHB protein product (Fig. 4D), suggesting that the impact of C136U editing on protein expression levels is subtle and quantitative.

Analysis of genomic DNA for C136T editing

To test whether the C136U RNA mutation has a corresponding C-to-T genomic DNA mutation in SDHB exon 2, we compared mutant RNA and DNA levels using our previously described semi-quantitative method based on PCR amplification and Taq1 restriction enzyme digestion (Baysal, 2007). We found no evidence of a corresponding DNA mutation in the hypoxic samples containing high levels of C136U RNA mutations (Fig. 4C). These results indicate that C136U mRNA editing occurs during or after transcription.

High throughput sequencing of SDHB and SDHD transcripts

To obtain a global view of transcript editing in the SDHB and the SDHD genes, we performed high throughput sequencing of the full length coding transcripts obtained by RT and high-fidelity PCR. We sequenced the four pairs of normoxic cultures that were characterized by microarray gene expression analyses. Each pair derived from the same PBMC and contained a sample with a low C136U editing rate (day 3 culture) and a sample with a high C136U editing rate (day 5–8 culture). Interrogation of the transcript sequences confirmed C136U mutations in each of the high-editing-rate sample. On average, 40% higher editing levels were seen by high throughput sequencing than those estimated by AS qPCR (Table 1). No additional canonical RNA edits in the form of A-to-I/G or C-to-U/T changes were identified in the SDHB transcripts. No non-canonical SDHB transcript variants could be confirmed by Sanger sequencing. In contrast, Sanger sequencing confirmed C136U in all four samples that showed high editing rates by AS qPCR and by high-throughput sequencing (Fig. S1E). Similarly, SDHD transcripts showed no evidence of RNA editing by high-throughput sequencing. Taken together, these results demonstrate that SDHB C136U editing is not accompanied by other transcript variants either in SDHB or SDHD and suggest that a site-specific C-to-U RNA editing mechanism regulates the SDHB functional transcript dosage.

Analysis of transcriptome sequence databases

We examined RNA sequencing data of the Illumina Human Body Map and ENCODE Human Transcriptome projects (Derrien et al., 2012; Tilgner et al., 2012; Brazma et al., 2003) for sequence variations in the ORF region SDHB mRNA. As expected, among the 16 normal human tissues from Human Body Map, the highest levels of SDHB C136U editing was seen in white blood cells (1.7%), with the other 15 tissues showing an average editing level of 0.2% (SD = 0.3%), with ovary (1%) and kidney (0.9%) being those with the highest levels among the 15 (Table S5). Among the 45 samples of 23 different cell-types of the ENCODE set, the editing rate was highest for the two primary CD14+ monocytes (1.9% and 2.6%), with the other samples showing an average 0.1% (SD = 0.2%) editing rate (Table S6). The editing rates noted for the samples for white blood cells and the monocytes are unlikely to be a result of sequencing error since a base call of U instead of C was made in only 0.04%–0.09% of the reads when all 213 C-bearing positions of the SDHB coding region were examined (Chi square test, P < 0.001). These results support our earlier conclusions that C136U editing occurs at low but statistically significant levels in peripheral blood monocytes.

Conclusions

We found that an acquired RNA nonsense mutation (R46X) introduced by C-to-U RNA editing dynamically reduces the SDHB functional transcript dosage in monocytes during early macrophage differentiation in normoxia and upon short-term exposure to hypoxia. The fraction of C136U transcripts was variable in monocyte-enriched PBMC cultures and peaked around 19% in normoxic cultures but increased to up to 49% within 2 days in hypoxia. The editing rates must be higher in monocytes because CD14+ cell isolation increased the percentages of C136U transcripts by 1.25-fold in normoxic cultures and 1.68-fold in hypoxic cultures. In contrast, no other cell type showed convincing evidence of SDHB RNA editing. Cytokine-driven differentiation of monocytes to macrophages or dendritic cells, or LPS stimulation of monocytes reduced the C136U editing rates under normoxia. Gene expression analyses identified CDA as a candidate gene for SDHB editing. To our knowledge, these results provide the first examples of hypoxia-inducible coding RNA editing and a programmed mutation targeting an endogenous nuclear gene in myeloid cells.

On the basis of genetic studies on SDH-mutated paragangliomas that show constitutive activation of hypoxia-inducible pathways, we hypothesize that reduction in the SDHB dosage by RNA editing facilitates hypoxia adaptation in monocytes by amplifying hypoxia signaling from mitochondria. The programmed SDHB RNA editing during normoxic macrophage differentiation may serve to augment signaling of inflammatory hypoxia. Some parallels can be drawn between the monocytes circulating in the high-oxygen environment of the peripheral blood and subsequently migrating into the hypoxic areas where differentiation into macrophages occurs versus certain facultatively aerobic organisms such as the intestinal parasitic worm Ascaris suum (A. suum). A. suum spores living in open air are exposed to high oxygen concentrations and utilize SDH; whereas parasitic adult worms living in the hypoxic environment of intestine use fumarate reductase (Kita et al., 2002). Thus, suppression of SDH might contribute to increased monocyte/macrophage survival in hypoxia. Because SDHB protein levels do not appear be altered significantly, it is conceivable that other hypoxia-inducible changes may also contribute to enhanced glycolysis observed in hypoxic monocytes (Roiniotis et al., 2009).

The mechanisms linking hypoxia sensing and signaling to SDHB RNA editing remain to be determined. It is conceivable that HIF transcriptional activity is required for induction of one or more components of putative cytidine deaminating RNA editor. Whether hypoxia-sensing leading to SDHB RNA editing in monocytes originates in the oxygen-dependent prolyl hydroxylase enzymes that post-translationally regulate hypoxic stabilization of HIFs also remains to be confirmed (Kaelin & Ratcliffe, 2008). The method described here will allow exploration of these mechanistic questions.

More broadly, our results extend previous observations associating external factors with differential A-to-I RNA editing (Garrett & Rosenthal, 2012; Sanjana et al., 2012; Balik et al., 2013) and demonstrate in a robust experimental model that coding RNA editing of certain genes occurs dynamically in a cell type and environment-dependent manner. Our findings suggest that gene targets of RNA editing can be missed by whole transcriptome sequence analyses unless tissues are examined in their physiologically relevant states. If the threshold for significant RNA editing is set at 10% or more in whole transcriptome sequence analyses (for example, Park et al., 2012), the SDHB gene would be identified as a target of RNA editing in monocytes upon exposure to hypoxia or during macrophage differentiation in normoxia but not in peripheral blood.

If SDHB editing aberrantly occurs in the paraganglionic tissues, an increased risk of paraganglioma development might ensue. Recent evidence suggests that tumor suppressor gene functions can be compromised by as little as a 20% decrease in gene dosage (Berger, Knudson & Pandolfi, 2011). Dosage sensitivity of the SDH complex is specifically supported by the imprinted transmission of PGL tumors by SDHD mutations as paternal but not maternal transmission of SDHD mutations is known to cause highly penetrant tumors. Recent discovery of tissue-specific imprint marks in the vicinity of SDHD suggests that subtle allelic expression differences can result in discordant penetrances depending on which allele carries the mutation (Baysal et al., 2011). If programmed downregulation of SDHB by RNA editing improves monocyte survival in hypoxic environments, therapeutic manipulation of this pathway might provide a tool to modify risk in common diseases associated with macrophage infiltration.

Supplemental Information

Figure S1 Miscellaneous supplementary figures

(A) Impact of long-term culture and cell density on C136U editing are shown. Five monocyte-enriched PBMCs were cultured long-term and the C136U editing rates were measured. The editing rate was higher in adherent cells on days 5–7 than in the uncultured monocyte-enriched PBMCs and on days 12–14 and 19–21 (P = 0.017, n = 5, one-way ANOVA). (B) Maximum C136U editing rate in the adherent cells shows very weakly positive correlation with the total number of monocyte-enriched PBMCs per well. Total number of samples = 31. (C) Attached aggregates (AAs) on culture day 20 are shown. The AAs start loosening after one week in culture and eventually appear as flat areas of increased cellular density. This culture was fed with fresh RPMI-1640/10% FBS on day 8. (D) Cytokine mediated differentiation towards macrophages or dendritic cells reduce C136U editing rates. Cultures containing 10% FCS showed higher mutation rates than those treated with M-CSF (p = 0.012) or GM-CSF/IL4 (p = 0.04). Multi-day averages from two PBMC cultures are shown. Matched samples were collected on days 4–8 for the m-CSF culture (n = 8) and days 4–6 for the GM-CSF/IL4 cultures (n = 5). The editing rate was obtained from adherent cells in 10% FCS and m-CSF treated (macrophage-differentiation) wells and from non-adherent cells in supernatant in GM-CSF/IL4-treated (dendritic cell-differentiation) wells. (E) Sanger sequencing confirms C136U RNA editing in all four samples that showed high editing rates by RT-qPCR and high-throughput sequencing. Chromatograms show examples of a sample with a low editing rate (sample 1, day 3 in Table 1) and one with a high editing rate (sample 4, day 7 in Table 1). C136U variant is shown by arrows.

Click here for additional data file.

Table S1 Oligonucleotide primers and probe sequences

Click here for additional data file.

Table S2 Efficiency and specificity of Total and T assays in determining C136U transcript levels

Click here for additional data file.

Table S3 Differentially expressed genes (>3-fold) accompanying increased SDHB RNA editing rate in normoxia

Click here for additional data file.

Table S4 Gene Ontology categories that show significant expressional changes during increased SDHB editing in normoxia

Click here for additional data file.

Table S5 Transcript sequence variation at position 136 of SDHB mRNA in BodyMap study samples

Click here for additional data file.

Table S6 Transcript sequence variation at position 136 of SDHB mRNA in ENCODE study samples

Click here for additional data file.

Video S1 Time-lapse video showing formation of monocyte-rich adherent aggregates in culture

Culture is initiated after mild monocyte enrichment by cold aggregation. After 5 h the supernatant is removed and hourly photographic imaging is started. Video is a composite of sequential images (time-lapse) for the first 48 h. Note that the aggregates start as a loose collection of cells and become larger and more compact by cell recruitment.

Click here for additional data file.

We thank anonymous platelet donors, personnel at RPCI’s core facilities for assistance with flow cytometry, high-throughput sequencing, immunocytochemistry and hypoxia treatment, Eric Kannisto for technical help, and the Department of Pathology and Laboratory Medicine for administrative support.

Additional Information and Declarations

Competing Interests

Author Contributions

Human Ethics

Microarray Data Deposition

The authors declare no conflict of interest.

Bora E. Baysal conceived and designed the experiments, performed the experiments, analyzed the data, wrote the paper.

Kitty De Jong and Biao Liu performed the experiments, analyzed the data, wrote the paper.

Jianmin Wang and Paul K. Wallace conceived and designed the experiments, analyzed the data, wrote the paper.

Santosh K. Patnaik analyzed the data, wrote the paper, analyzed Illumina and ENCODE databases for SDHB RNA editing.

Robert T. Taggart conceived and designed the experiments, performed the experiments, analyzed the data.

The following information was supplied relating to ethical approvals (i.e., approving body and any reference numbers):

This study was approved as exempt by the IRB.

NHR 025712: “Epigenetic regulation of mitochondrial complex II by SDHB editing”.

The following information was supplied regarding the deposition of microarray data:

Microarray data have been deposited in GEO (www.ncbi.nlm.nih.gov/geo/) under accession number GSE45900.

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
