# Peer review of "Hypoxia-inducible C-to-U coding RNA editing downregulates SDHB in monocytes"

_PeerJ, doi:10.7717/peerj.152_

## Round 0.1 · original submission · Major Revisions

Dear Authors,

I believe that your paper is worthy of publication but please try to especially address the concerns of the first reviewer. I am not concerned about comment 3 but I do believe that a response to the first 2 concerns is needed.

·

Basic reporting

See below

Experimental design

See below

Validity of the findings

See below

Additional comments

The manuscript by Baysal et al studies the effect of hypoxia on C to U RNA editing of SDHB gene in monocytes. Previous studies show that SDH is a heterotetrameric tumor suppressor complex mutated in paraganglioma tumors due to hypoxia-inducible activation pathways. The authors studied regulation, extent and cell type origin of SDHB RNA editing. The authors found that RNA editing is low in monocyte-enriched PBMCs under normal condition, however, markedly increased under hypoxic conditions. This leads to down regulation of SDHB gene and consequently an increase in monocyte cell survival. Overall these findings add the mechanistic details to our understanding of the RNA editing under physiologically relevant conditions such as hypoxia, however additional data would support and strengthen their conclusions.

Major concerns:

1. The authors show that hypoxia, one of the physiological conditions, increased the C136U RNA editing of SDHB transcripts in CD14+ monocytes.
- Are HIF-factors (HIF-1alpha/HIF-2alpha) involved in this process?
- Does the knockdown of these would rescue the change in the C136U RNA editing of SDHB transcripts in CD14+ monocytes?
- Is the RNA editing pattern changed when these CD14+ monocytes are treated with inflammatory conditions (for example LPS) which is also one of the microenvironmental conditions.
-
2. On page 22, the authors state that “the suppression of SDH might be an evolutionarily conserved metabolic adaptation to hypoxia….. due to enhanced glycolysis”. Though the data suggest that the C136U RNA editing is increased which should supposedly decrease the express, however, SDHB protein levels were unchanged under hypoxic conditions compared to normoxia (Figure 4D). I do not think that changes in the RNA editing of this gene is sufficient to conclude that SDHB expression is downregulated leading to enhanced glycolysis.

3. The authors showed RNA editing pattern in peripheral blood mononuclear cells (lymphocytes and monocyte-enriched population). Did the authors by chance look at the status of RNA editing in polymorphonuclear cells, especially neutrophils, which are also part of the immune system and have key role(s) in the immunological processes.

Minor issues:
1. Provide p values for the figures: 1C-E, 2C and 4A, B and E
2. Figure 4 legends, line 8, change C13U into C136U RNA editing

·

Basic reporting

1. In line 322, “>= fold change”, a “3” is missing.
2. In Fig.1A, 9 lymphoblastoid cell lines and one fibroblastic cell line were included, but the authors did not provide names and sources of these cell lines.

Experimental design

No Comments

Validity of the findings

No Comments

Additional comments

In this manuscript, Baysal investigated RNA editing of SDHB (C136U) in different cell types and human tissues using both bioinformatics tools and wet lab tools. They provided massive amount of data to support their hypothesis that hypoxia induces RNA editing of SDHB and downregulates its expression in monocytes. The authors have performed many hypothesis-driven and well-designed experiments. Overall, this manuscript is well-written and the results are properly interpreted with some speculations. Two minor points are listed below:
1. In line 322, “>= fold change”, a “3” is missing.
2. In Fig.1A, 9 lymphoblastoid cell lines and one fibroblastic cell line were included, but the authors did not provide names and sources of these cell lines.

---

## Round 0.2 · accepted · Accept

You changes were accepted and the manuscript is now ready for publication. Congratulations.